# Detecting Quantum Critical Points of Correlated Systems by Quantum Convolutional Neural Network Using Data from Variational Quantum Eigensolver

**Nathaniel Wrobel** [1,2]**, Anshumitra Baul** [2]**, Ka-Ming Tam** [1,2,*] **and Juana Moreno** [1,2,*]

1    Center for Computation & Technology, Louisiana State University, Baton Rouge, LA 70803, USA
2    Department of Physics and Astronomy, Louisiana State University, Baton Rouge, LA 70803, USA
*    Correspondence: phy.kaming@gmail.com (K.-M.T.); moreno@lsu.edu (J.M.)

**Abstract:** Machine learning has been applied to a wide variety of models, from classical statistical mechanics to quantum strongly correlated systems, for classifying phase transitions. The recently proposed quantum convolutional neural network (QCNN) provides a new framework for using quantum circuits instead of classical neural networks as the backbone of classification methods. We present the results from training the QCNN by the wavefunctions of the variational quantum eigensolver for the one-dimensional transverse field Ising model (TFIM). We demonstrate that the QCNN identifies wavefunctions corresponding to the paramagnetic and ferromagnetic phases of the TFIM with reasonable accuracy. The QCNN can be trained to predict the corresponding 'phase' of wavefunctions around the putative quantum critical point even though it is trained by wavefunctions far away. The paper provides a basis for exploiting the QCNN to identify the quantum critical point.

**Keywords:** quantum neural network; convolutional neural network; quantum phase transition; transverse field Ising model; quantum convolutional neural network; variational quantum eigensolver; variational method; quantum computing; quantum machine learning

## 1. Introduction

Machine learning (ML) and quantum computing (QC) are among the most notable topics that significantly impact various fields of physics. ML has become a powerful tool for scientific and academic use in the age of big data. The progress of QC, in particular, the realization of quantum computers with tens of qubits, may provide a new opportunity to study challenging problems in strongly correlated many-body physics, among other applications.

The motivation for the present work was to take advantage of recent developments in quantum algorithms to find the ground state of the many-body Hamiltonian and classify quantum states [1–5]. Understanding quantum criticality is the driving force of many exotic phenomena in condensed matter physics and material science [6,7]. In particular, the theory to explain non-Fermi liquid is based on the existence of a quantum critical point in high-temperature superconducting cuprates [6,8]. Unfortunately, numerical studies are relatively limited, primarily due to the minus sign problem in the quantum Monte Carlo algorithm. Thus, a new direction for studying quantum critical points may be essential for analyzing strongly correlated systems.

ML has been applied to physics and other branches of science and engineering. Explosive growth has been seen in diverse applications in the past decade or so. This growth is principally driven by the availability of an extensive dataset and accessible libraries for sophisticated deep learning methods based on neural networks [9,10]. Among the different types of neural network, the convolution neural network (CNN) is widely used [11]. Unlike conventional dense or fully connected neural networks, CNN emphasizes local correlation information. It serves as a high-performance classifier for computer vision. Image identification is a central topic for classifiers. Most images have a certain level of spatial correlation.

CNN is designed to utilize the local spatial correlations in the input data. In practice, most physical data also possesses a robust spatial correlation; therefore, in hindsight, it is not surprising that CNN has seen many successful applications in physics.

In ML, CNN has been adopted mainly to identify phase transitions for classical statistical models from snapshots of classical Monte Carlo or molecular dynamics configurations and also configurations from quantum Monte Carlo of strongly correlated systems [12–16]. Attempts have been made to use the quantum wavefunction from exact diagonalization [17–29]. Recent studies further involve feeding spatially resolved experimental data from scanning tunneling microscopy to identify different phases of materials [30].

However, a quantum computer for fault-tolerant quantum computations, which can supersede the best classical computer for many tasks, may not be available in the near future. Noisy quantum computers with tens of qubits are immediately available. Noisy intermediate-scale quantum (NISQ) computers are likely to be feasible shortly [31]. They open up new opportunities to use quantum computation to solve problems strikingly different from classical numerical simulations. Among the various methods feasible on such NISQ computers, the variational quantum eigensolver [32] and the general idea of the quantum approximation optimization algorithm represent promising proposals [33].

An enormous amount of effort for addressing problems in optimization, chemistry, and strongly correlated systems has been invested in recent years [3,34]. Conceptually, the approach is based on a quantum state with parameters. The quantum computer is used to calculate the expectation value of a given quantum state to the quantity produced for optimization [34]. This can be a cost function, an optimization problem, or the ground state energy of a molecule. In general, calculating such an expectation value scales exponentially to the problem size by classical methods. The quantum computer offers an opportunity to speed up such calculations. A classical optimization algorithm then optimizes the parameter. The idea of variational methods is not limited to ground state calculation; it is a general concept used to mimic any operator in the variational sense. For example, quantum dynamics based on solving the Schrödinger equation can be estimated by the variational method [35,36].

Variational methods have been widely adopted in condensed matter physics. Specifically, the variational quantum Monte Carlo (VMC) is one of the effective numerical methods for solving correlated systems [37,38,38–42]. Monte Carlo calculates the quantum expectation values for the ground state energy. The multivariate minimization method minimizes the ground-state energy with respect to the variational parameters of the wavefunction. Its main advantage is the absence of the minus sign problem, which hinders most quantum Monte Carlo methods for fermion problems.

The VQE provides a new framework for sidestepping the computational intensive part of the conventional VMC method in calculating quantum expectation values by quantum computers [34]. The wavefunctions represented in quantum circuits also provide new opportunities and challenges due to the different nature of the wavefunctions used in the conventional VMC [43]. It is worth noting that most numerical methods for finding the 'ground state' of a many-body system are based on the non-unitary propagation of a trial state; a typical example is the projection quantum Monte Carlo [44].

From the viewpoint of utilizing quantum computing approaches for strongly correlated systems, the ground state energy calculation alone is often insufficient to reveal much detail of the system. An exciting issue is the possibility of quantum phase transitions at zero temperature by tuning the parameters in the Hamiltonian [7,45]. The ground-state energy of relatively small system sites, which could be simulated in the near future, does not provide a direct answer to determine a quantum phase transition. Constructing an order parameter corresponding to the known broken symmetry in the thermodynamic limit allows direct access to phase transitions. Given the small system size and the nature of a second-order phase transition of quantum phase transitions, an order parameter alone is often a more obscure way to tell whether the systems possess a phase transition. A true singularity at

the phase transition is realized only in the thermodynamic limit. This challenge has led to the development of the finite-size scaling method [46]. However, proper finite-size scaling may not be feasible for quite small system sizes that can be simulated. Moreover, there are systems where phase transitions do not appear in local-order parameters, or the order parameter is simply unknown [47,48].

The notion of using a quantum circuit as a classifier or clustering algorithm began to attract attention more than two decades ago [49–55]. The concept of the classical neural network has been filtered into the idea of quantum classifiers in recent years [1,56–77]. The classical neural network consists of links and neuron units represented by activation functions, which are organized in a layered structure. The key properties of CNN are translationally invariant convolution and pooling layers, each characterized by a constant number of parameters (independent of the system size), and sequential data size reduction (i.e., a hierarchical structure) [78].

Using a neural network as a classifier decreases the number of neuron units in each layer until one or a few units remain in the output layer. Thus, each layer can be regarded as a pooling layer because the number of inputs to the neuron units at each layer is smaller than the number of outputs. It is a method for compressing and reducing the degrees of freedom; thus, the suggestion of a renormalization group can be pertinent in particular neural networks.

For most space-dependent data, there is a non-zero spatial correlation. Short-range correlations appear in data from images of objects to physical systems, such as spin correlation and spatial correlation of the positions of atoms or molecules in a solid phase and even in a liquid phase. Therefore, it is not surprising that CNN has seen many applications in physics in learning patterns from statistical models to strongly correlated systems.

CNN is realized by introducing a so-called convolutional layer within each layer of activation functions of a dense neural network. The purpose is to extract 'hidden' information through some combination of local data, which is missing in the standard dense neural network. In practice, the combination is a weighted sum of local data.

A simple analogue can be drawn to the quantum circuit by replacing the links and activation by the quantum links and the quantum gates, respectively [78]. The principal structure of a QCNN is composed of two distinct types of layers. First, the pooling layer, which reduces the degrees of freedom, can be replaced by multi-qubit gates. The simplest possibility is the CNOT gate [78]. Second, the convolution layer in the CNN can be replaced by multi-qubit quantum gates among nearby qubits. Thus, QCNN can be understood naively as a quantum neural network classifier with convolutional layers in which 'convolution' between nearby qubits can be processed.

In short, a quantum circuit model is introduced, which extends the key properties of the classical CNN to the quantum domain. The circuit's input is a quantum state. A convolution layer applies a single quasilocal unitary in a translationally invariant way for finite depth. A fraction of qubits are measured for pooling; their outcomes determine unitary rotations, which are applied to nearby qubits [78]. Hence, the non-linearities in QCNN arise from reducing the number of degrees of freedom. Convolution and pooling layers are performed until the system size is sufficiently small. Then, a fully connected layer is applied as a unitary function on the remaining qubits if needed. The outcome of the circuit is finally obtained by measuring a fixed number of output qubits. Similarly, in the classical CNN, circuit structures (i.e., hyperparameters of QCNN), such as the number of convolution and pooling layers, are fixed.

Recent studies have shown that quantum-enhanced machine learning is a promising approach for recognizing the phase of matter [79]. An interesting question is whether the QCNN method can identify different phases of a quantum many-body system. This is the first step towards applying it for detecting quantum phase transitions. Using ML to identify phases by inputting the wavefunction is a challenge, as the Hilbert space of the system increases exponentially with respect to the system size. A practical method to bypass such a challenge is to consider the reduced density matrix or some other derived quantities based on the wavefunction [18].

An evident advantage of the QCNN approach is that the input is naturally quantum mechanical; the wavefunction does not need to be written as a classical vector. The dimension grows exponentially to the system size to be fed to the ML method. The disadvantage, or perhaps an unknown factor, is that the wavefunction is not calculated exactly, and that there is no control parameter to systematically improve the wavefunction.

It has to be input as some form of a quantum circuit—the one which is most promising in the NISQ is the VQE. The primary purpose of the present paper is to present a study of a many-body quantum system solved by VQE and then to use the QCNN to identify the VQE wavefunction corresponding to the different phases of the model. This provides a possible framework for extracting quantum critical points.

The use of quantum algorithms for machine learning is a rapidly developing topic. Some of the latest developments have included, but have not been limited to, edge detection [80], quantum particle swarm optimization [81], quantum circuit Born machine [82], and image generation via generative network [83]. We refer readers interested in the proposed applications to a recent review paper [84].

This paper is organized as follows: In Section 2, we briefly describe the transverse field Ising model (TFIM). In Section 3, the data from the VQE of the TFIM is discussed, and the structure of the QCNN is presented. The results from the variational autoencoder are described in Section 4. We conclude the paper and discuss the implications and possible future applications of the method developed in this study in Section 5.

## 2. Transverse Field Ising Model

### 2.1. Model

We consider the one-dimensional Ising model with a transverse field. The Hamiltonian is given as

$$H = -J \sum_{i=1}^{N} \hat{\sigma}_i^z \hat{\sigma}_{i+1}^z - \Gamma \sum_{i=1}^{N} \hat{\sigma}_i^x, \tag{1}$$

where $\hat{\sigma}^\alpha (\alpha = x, y, z)$ are the Pauli matrices that obey the commutation relation, $[\hat{\sigma}_i^\alpha, \hat{\sigma}_j^\beta] = 2\iota \delta_{ij} \epsilon_{\alpha\beta\gamma} \hat{\sigma}_i^\gamma$, where $\iota$ is an imaginary number. $J$ is the coupling between the nearest-neighbor spins and is set to 1 to serve as the energy scale of the problem. We only consider a ferromagnetic case with periodic boundary conditions.

$\hat{\sigma}^z$ has the eigenvalues of $\pm 1$, and their corresponding eigenvectors are symbolically denoted by

$$| \uparrow > = \begin{pmatrix} 1 \\ 0 \end{pmatrix} \tag{2}$$

and

$$| \downarrow > = \begin{pmatrix} 0 \\ 1 \end{pmatrix}. \tag{3}$$

The model is solvable in the sense that the eigenenergy can be obtained exactly via the Jordan–Wigner transformation. The quantum critical point can also be determined exactly by mapping the model to an anisotropic two-dimensional Ising model in a square lattice and employing the self-duality property of the model. The quantum critical point of the TFIM is at $\Gamma_c = J$ [85]. Given the relative simplicity of the model and that the value of the transverse field is exactly known at the quantum critical point, the TFIM provides a good test bed for the capability of a quantum classifier for identifying the phase transition of a quantum many-body system.

### 2.2. Wavefunction from VQE

As our goal is to demonstrate that the QCNN can identify the wavefunction in different phases, the input should be represented in a quantum circuit. It is possible to cast the wavefunction in terms of a classical vector into quantum data. This is precisely what needs to be done using a quantum classifier for classical data, such as identifying classical images.

We use the database provided by Tensorflow Quantum for TFIM [10,86]. There are, in total, 81 datapoints for $\Gamma = [0.2, 1.8]$ in the spacing of 0.02. The variational wavefunction is presented in the Figure 1. Each qubit in the wavefunction represents a spin in the TFIM. The first layer of the quantum circuit is composed of a Hadamard transform by adding a Hadamard gate to each qubit. Then, $N/2$ layers of gates are enacted on the quantum circuit. Each layer contains two sub-layers. The first sublayer comprises $N$ ZZ Ising coupling gates, which act on each pair of nearest-neighbor spins represented by the corresponding qubits. The rotation angle is fixed for each ZZ Ising coupling gate within the same sublayer. The second sublayer is composed of $N$ Pauli X gates, each acting on a qubit with a fixed rotation angle within the sublayer. Therefore, the quantum circuit contains $N/2 \times (1 + 1) = N$ variational parameters.

Only 81 datapoints are available in the Tensorflow Quantum database, which is a relatively small number for applying ML [86,87]. We generate additional datapoints for finer grids of $\Gamma$ points so that more data is available for the training and testing of the QCNN. We did not optimize as in the standard VQE method to obtain the new data. We take advantage of the fact that all the variational parameters vary smoothly as a function of the transverse field $\Gamma$ in the TFIM. The additional datapoints are simply generated by linear interpolation of the variational parameters with respect to $\Gamma$.

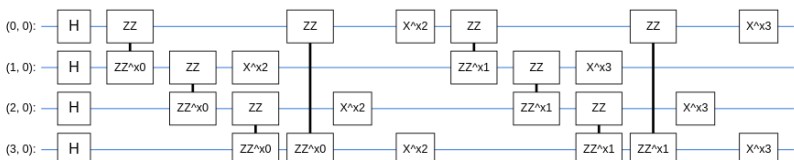

**Figure 1.** Variational wavefunction for the TFIM in the Tensorflow Quantum database, for $N = 4$ [86]. The notation of the quantum gates is adopted from the Cirq [87]. H is the Hadamard gate, X is the Pauli X gate with the rotation angle associated with the number, and ZZ is the Ising coupling gate with the rotation angle given by the associated number. An Ising coupling rotation is given explicitly as $ZZ^\wedge \theta = exp(-i\frac{\theta}{2} Z \otimes Z)$ for the rotation along the z direction, and similarly for the rotations along the x and y directions. $x0, x1, x2, x3$ are the four parameters for the wavefunction.

We note that the choice of a variational wavefunction is not unique. Choosing an optimized variational wavefunction has been an important, but largely unsolved, problem, even in the context of the classical variational method as routinely performed by VMC [38]. We do not attempt to find the 'best' possible wavefunction in the present study. The 'best' wavefunction is model- and even parameter-dependent. We fix the functional form of the wavefunction for the entire range of transverse field strength, which likely is not the 'best' optimized wavefunction.

### 3. QCNN

By generating a larger set of datapoints, we can train a QCNN. The input for our QCNN model is a quantum circuit. This quantum circuit represents our wavefunction from the VQE solver for the TFIM.

The QCNN is another quantum circuit constructed with an alternating series of convolutional and pooling layers until the number of pooling layers dwindles to a single qubit. The idea is that the quantum circuit can acquire important information from the input quantum data. An example of a four qubit QCNN is shown in the Figure 2.

An $N$ qubit input requires $N/2$ convolutional units and $N/2$ pooling units to reduce to $N/2$ output qubits. Therefore, for the input of $N$ qubits, assuming that $N$ is a power of two, the QCNN circuit contains $\sum_{i=1}^{logN/log2} 2^{i-1}$ convolutional units and pooling units. The $N$ qubits reduced to a single qubit are then measured as the output.

For the detection of the quantum critical point of the TFIM, the goal is to prepend our datapoints to a QCNN model and train it to identify the 'phase' of each wavefunction correctly. This is performed in a supervised environment, as we already have the correct

phase identification for each datapoint in the database. We have either a ferromagnetic or paramagnetic phase for a given transverse field. In the $\Gamma = [0.2, 1.8]$ range, the system is in a ferromagnetic phase below 1 and a paramagnetic phase above 1.

The pooling unit we used for the present study, which pools two qubits into one qubit, is shown in the Figure 3. It consists of local rotation gates on each qubit, a controlled X-gate, and the inverse rotation on the controlled bit. Thus, in total, there are six parameters in each pooling unit that pool two qubits into one qubit.

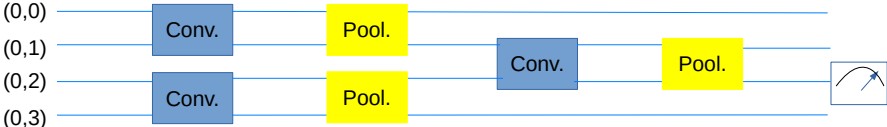

**Figure 2.** An example of the entire QCNN circuit for 4 qubits, two convolutional units act on the pairs of nearest neighbor qubits. A pooling unit which reduces the 4 input qubits into 2 output qubits. Another layer of convolutional units on the nearest neighbor pair of qubits, and the pair of qubits is then fed into another pooling unit with one output qubit. The final qubit is measured. Note that the parameters in one convolutional unit can be different from the other, as may the pooling units.

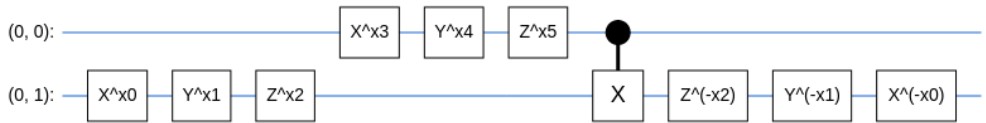

**Figure 3.** Pooling between two qubits. A pooling layer from $N$ qubits to $N/2$ qubit is composed of $N/2$ of pooling units. The circuit contains single qubit rotation gates and a CNOT gate. $x0, x1, \ldots, x5$ are the parameters.

The convolutional layer is composed of multiple two-qubit convolution units. A convolution unit for two qubits is shown in Figure 4. It consists of local rotation gates on each qubit sandwiched between the Ising coupling gates for the two qubits. Therefore, there are 15 parameters for each convolutional unit between two qubits.

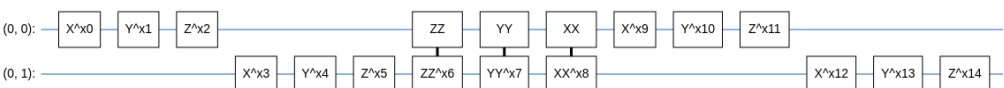

**Figure 4.** Convolution between two qubits. A convolutional layer of $N$ qubits is composed of $N/2$ convolutional units acting on the qubits which represent pairs of nearest neighbor spins. The notation of the quantum gates is adopted from the Cirq. The circuit contains single spin rotation gates and Ising coupling gates. XX, YY, and ZZ are the Ising coupling gate with the rotation angle given by the associated number along the x, y, and z directions, respectively. $x0, x1, \ldots, x14$ are the parameters.

In addition to single-spin rotations, the convolution between two qubits is performed via a series of rotation gates on the coupling in the x, y, and z directions. The pooling between two qubits is via a controlled X-gate. A natural question is how to guarantee that the designated convolutional units and pooling units provide the 'best' optimized neural-network-like structure to produce the 'best' performance.

As for the classical convolutional neural network, the 'best' network architecture is mainly obtained through trial and error. We do not attempt to find the 'best' architecture available. We set a simple enough architecture that is easily optimized but sufficient to provide reasonably good predictions.

The code for the QCNN is written in Cirq and uses the Tensorflow Quantum package for training [86,87].

## 4. Results

We trained the QCNN using two different sets of training data. For the first, the training of the QCNN was carried out randomly by picking 80% of the wavefunctions with labels to designate their corresponding phases as the training data set. For the second, only the datapoints related to the small and large transverse fields were used for training. We defined the label for the ferromagnetic phase as $-1$ and the paramagnetic phase as $+1$. For benchmarking the accuracy of predictions, we cast the output to $-1$ if the measurement of the output from the QCNN was smaller than 0, and similarly, cast it to $+1$ if the measurement was larger than 0.

### 4.1. Training QCNN with Data for Randomly Picked Data for $0.2 \leq \Gamma \leq 1.8$

The accuracy of the trained QCNN was benchmarked against the remaining 20% of the available samples. We show the loss and accuracy during each iteration of the training processes for three system sizes $N = 4, 8$, and 12.

By training our QCNN with randomly chosen wavefunctions, we allowed our network to study samples across the entire dataset. The QCNN becomes familiar with values of $\Gamma$ close and far from the quantum critical point. In doing so, we expected the QCNN to be familiar with the wavefunctions, giving us valid results for the trained data predictions. This is exactly what we observed when using a system size of $N = 4$ (see Figure 5). The accuracy for both the training and testing data was consistently high, and we observed minimal fluctuations in accuracy from start to finish. This method of randomized data allowed the QCNN to adjust to the variations in wavefunctions during training and then to apply this to our testing data. We observed very similar results for the accuracy with system sizes $N = 8$ and 12 (see Figures 6 and 7). Increasing the system size had no significant impact on the results for accuracy. The ability to predict the 'phase' of wavefunctions proved to be a task that our QCNN could execute.

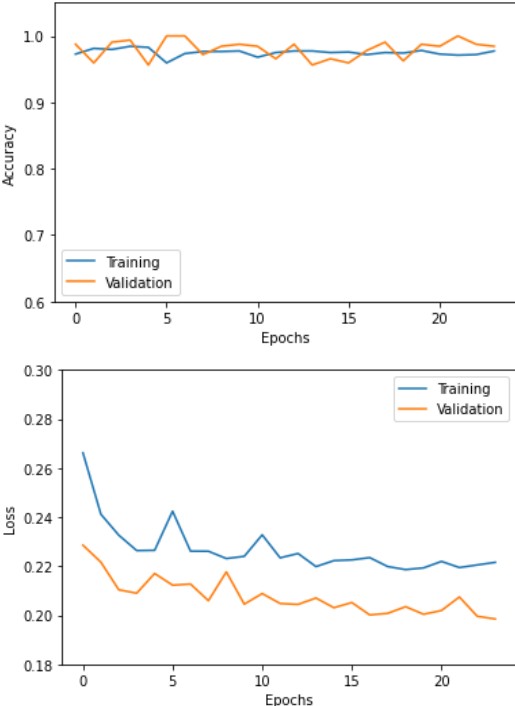

**Figure 5.** Accuracy and loss for the training and validation datasets of the QCNN for $N = 4$ as a function of the number of epochs.

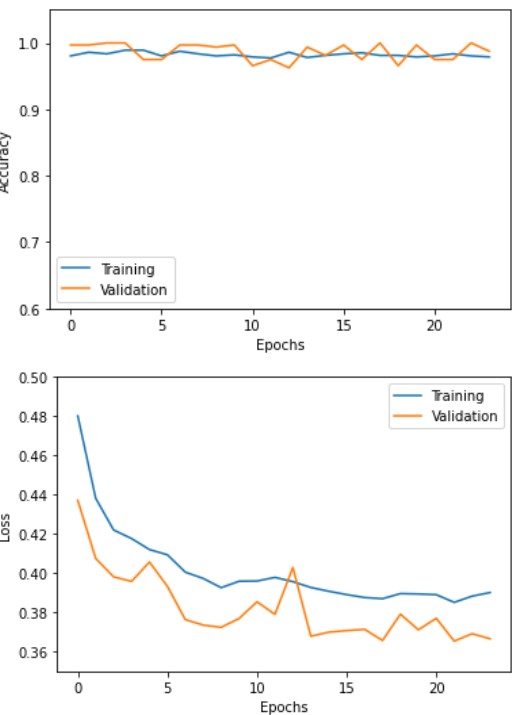

**Figure 6.** Accuracy and loss for the training and validation datasets of the QCNN for $N = 8$ as a function of the number of epochs.

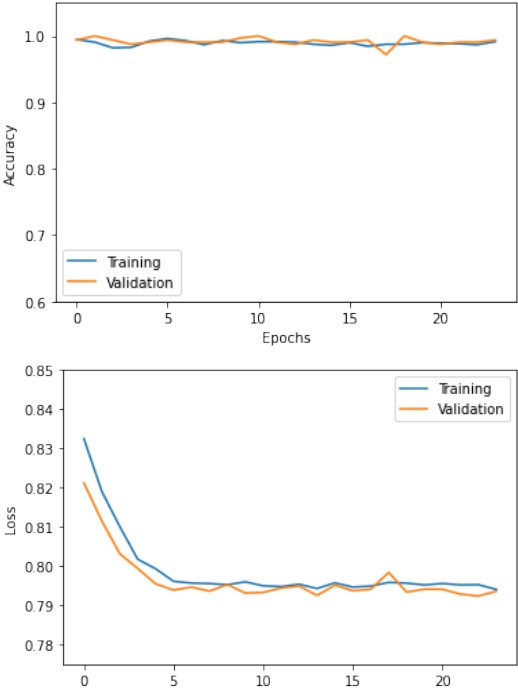

**Figure 7.** Accuracy and loss for the training and validation datasets of the QCNN for $N = 12$ as a function of the number of epochs.

We observed what happened as the QCNN underwent the training phase, studying the results of the network's loss. The network became fully trained once the loss converged. Once the loss stopped decreasing, the QCNN was fully trained and at its maximum potential. The loss of system size $N = 4$ decreased at a high rate during the initial training iterations and then began to flatten as the QCNN reached its full potential. We noticed

a pattern when observing larger system sizes. As more input variables were included, the initial and final loss increased. Regardless of the increased loss value, we were still able to observe how the system became trained at each iteration. All the system sizes tested allowed visualization of the QCNN's training phase and showed that the network could improve its ability to provide correct predictions for the 'phase' corresponding to the wavefunctions.

### 4.2. Training with Data for 0.5 < Γ and Γ > 1.5

We tested QCNN with the chosen training and testing data in light of the above results. For training, we used wavefunctions with values of Γ below 0.5 and above 1.5. The testing data were in the range of $[0.5, 1.5]$. This allowed observation of how the QCNN behaved when classifying data near the known quantum critical point ($\Gamma_c = 1$) after being trained with data far from the quantum critical point. One hundred percent accuracy during the training phase was observed for each system size. This was perhaps not surprising, given that the two datasets for very large and very small Γ were far apart.

In Figures 8–10, we plot the results for $N = 4$, 8, and 12, respectively. The accuracy from the testing data was consistently lower than that from a randomized dataset, but showed a sharp increase during its initial iterations. High accuracy in training indicates that the QCNN can classify datapoints far from the quantum critical point. As we approached the quantum critical point, the network had more difficulty with predictions. The inability to familiarize itself with datapoints in this range impaired the accuracy of the testing data. The sharp increase shows that, after encountering a few datapoints near quantum critical, the network can learn to improve its capability in our range of $[0.5, 1.5]$. The loss showed small changes during the training phase compared to using randomized data. Although we still observed a decrease, this decrease was minimal. The QCNN had little room for improvement when the testing data were isolated from the quantum critical point.

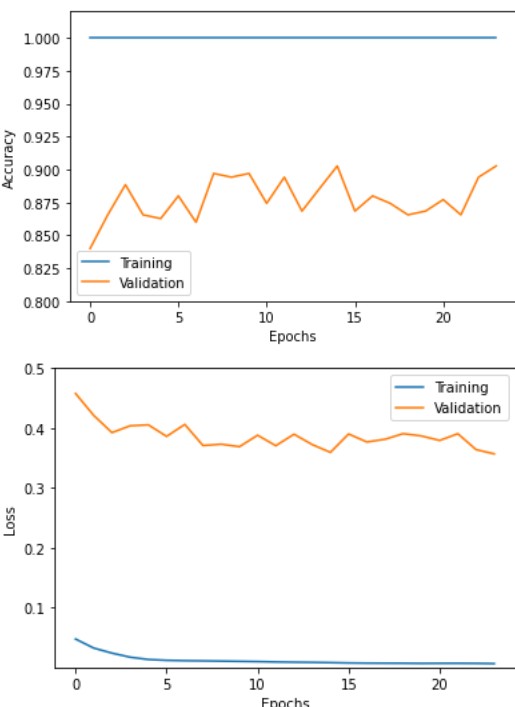

**Figure 8.** Accuracy and loss for the training and validation datasets of the QCNN for $N = 4$ with training data Γ < 0.5 and Γ > 1.5 as a function of the number of epochs.

The result of using only large and small values of Γ for training the QCNN is significant with respect to the prospect of using QCNN to detect the quantum critical point. A relatively

well-studied scheme for using supervised classical ML to detect phase transitions is to train a supervised classical ML, such as CNN, for the control parameters of a system. For example, the temperature in the thermal transition and the external parameter in the quantum phase transition are away from the putative phase transition or critical point [21]. The results demonstrate that the QCNN can be trained using data away from the putative quantum critical point and be used to predict the label of wavefunctions around the critical point with good accuracy.

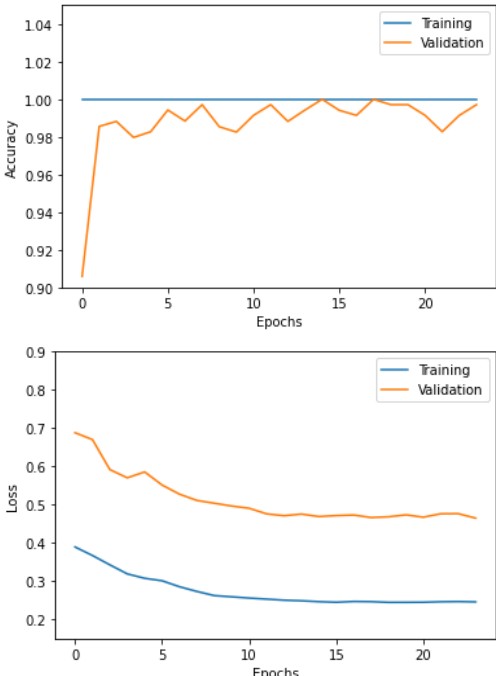

**Figure 9.** Accuracy and loss for the training and validation datasets of the QCNN for $N = 8$ with training data $\Gamma < 0.5$ and $\Gamma > 1.5$ as a function of the number of epochs.

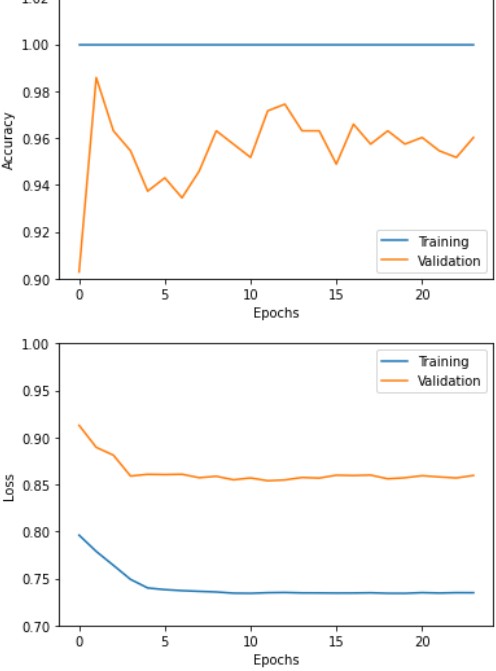

**Figure 10.** Accuracy and loss for the training and validation datasets of the QCNN for $N = 12$ with training data $\Gamma < 0.5$ and $\Gamma > 1.5$ as a function of the number of epochs.

### 4.3. Predicted Labels as a Function of Γ

To locate the quantum critical point, we can plot the predicted values from our QCNN to find a pattern in the outputs. In this calculation, we sought to find the value of Γ, where the QCNN predicted labels switch between the two phases, ferromagnetic and paramagnetic. This marks a good approximation to the quantum critical point. The quantum phase transition occurs at Γ = 1 for the TFIM, and we sought to observe this with the results given by the QCNN. A plot describing how the predicted labels change as Γ changes is needed to find if the network is capable of this. If the phase transition occurs at or near Γ = 1, then we can infer the validity of QCNN.

Observing these transitions for systems $N = 4, 8, 12$ in Figures 11–13, each trial holds a transition near Γ = 1. This deduces the value of Γ that holds a quantum critical point. The QCNN predicts a ferromagnetic phase for low values of Γ and a paramagnetic phase for high values of Γ. The network output for ferromagnetic prediction is for Γ < 1. As Γ approaches 1, we expect to find a quantum critical point. This can be identified by a sudden jump to a different output value, which signifies a paramagnetic phase. In the Figures 11–13, this jump is seen near Γ = 1. We observe a series of ferromagnetic predictions and then paramagnetic predictions. Our approximate quantum critical point is the value of Γ where the predicted phase change occurs. As our predicted quantum critical point consistently lies near the true value, we conclude that the network can be used to identify quantum critical points. It might be expected that the results for smaller system sizes would be more accurate and closer to the true critical point ($\Gamma_c = 1$). However, the wavefunctions for smaller system sizes may contain stronger fluctuations and, thus, adversely affect the accuracy of our approach.

Thus, the technology developed for classical supervised ML can be adapted for detecting quantum critical points [21]. The significant modification is to replace the classical supervised ML, such as the classical CNN for the classical data, with the QCNN for the quantum wavefunctions.

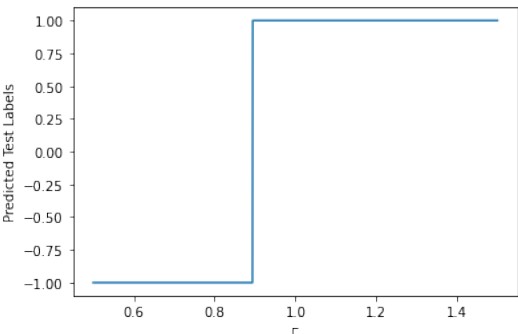

**Figure 11.** Predicted test labels vs Γ for $N = 4$. The predicted labels jump at Γ = 0.89.

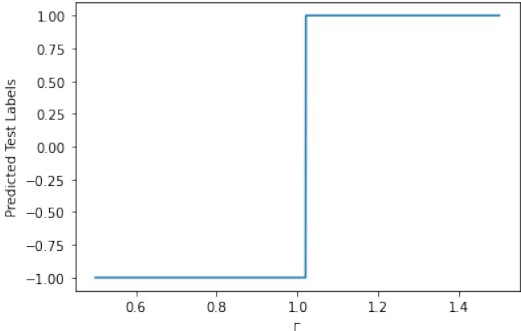

**Figure 12.** Predicted test labels vs Γ for $N = 8$. The predicted labels jump at Γ = 1.02.

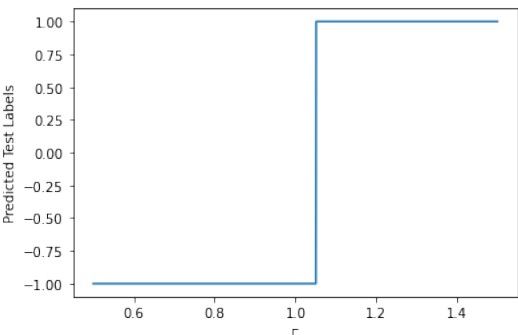

**Figure 13.** Predicted test labels vs $\Gamma$ for $N = 12$. The predicted labels jump at $\Gamma = 1.05$.

## 5. Discussion and Conclusions

We conclude from the results that the QCNN model can detect whether a TFIM wavefunction is ferromagnetic or paramagnetic. A QCNN model is trained to detect the represented phase of the wavefunction by implementing a dataset of varying external parameters. Once the training phase is completed, we will have a fully trained network that provides an efficient method for phase detection for the TFIM.

For multiple reasons, studying quantum phase transitions has been one of the major topics in condensed-matter physics. First, devising an effective theory for describing the quantum critical point is often complex. Various approaches have been proposed to understand the quantum critical point from a straightforward renormalization group from the upper critical dimension to more exotic gauge gravity duality approaches [7,45]. Second, the quantum critical point is responsible for many exotic behaviors in strongly correlated systems, including high-temperature superconductivity in cuprates [8], and non-Fermi liquids [6]. The study of the properties, or even the detection, of a quantum critical point, has been a significant topic in computational condensed matter physics for models from two to infinite dimensions [8,88–90].

The encouraging results of using QCNN to identify the ferromagnetic and the paramagnetic phase of the TFIM from the wavefunction obtained by VQE offers a new direction for the study of quantum critical points. This presents an opportunity that is different from that provided by other conventional computational approaches, such as quantum Monte Carlo and other diagonalization-based methods.

Whether such an approach can be applied to more exciting and complicated models, such as the Hubbard model beyond one dimension, requires to be addressed in future studies. There are two key questions that need to be answered. First, can the VQE provide a sufficiently accurate wavefunction for strongly correlated problems [91]? Second, can the wavefunction contain sufficient features that some form of quantum classifier can identify? These questions are more acute due to the limited system size of the models, which can be simulated on the NISQ machines. The ground-state energy alone or even the order parameters may not be beneficial for identifying phase transitions for small system sizes.

The nature of the Hamiltonian in chemistry problems is somewhat different from that of strongly correlated systems. The Hamiltonian for molecules is often diagonally dominated because the off-diagonal matrix elements are smaller than the diagonal ones. On the other hand, models for strongly correlated systems often have comparable off-diagonal and diagonal matrix elements. This is why some highly successful methods in computational chemistry, such as conventional coupled cluster theory, have seen limited success for strongly correlated systems. Therefore, whether the current approach can be transplanted to strongly correlated systems, which are relevant to a plethora of exotic properties of materials, remains to be examined.

**Author Contributions:** Conceptualization, N.W., A.B. and K.-M.T.; Methodology, N.W. and K.-M.T.; Data curation, N.W.; Writing—original draft, N.W., A.B. and K.-M.T.; Writing—review & editing, N.W., A.B., K.-M.T. and J.M.; Supervision, K.-M.T. and J.M. All authors have read and agreed to the published version of the manuscript.

**Funding:** This manuscript is based on work supported by NSF DMR-1728457. This work involved use of the high-performance computational resources provided by the Louisiana Optical Network Initiative (http://www.loni.org (accessed on 1 December 2022)) and by HPC@LSU computing. JM and KMT are partially supported by the U.S. Department of Energy, Office of Science, Office of Basic Energy Sciences under Award Number DE-SC0017861. NW is supported by NSF OAC-1852454 with additional support from the Center for Computation & Technology at Louisiana State University.

**Data Availability Statement:** Not applicable.

**Conflicts of Interest:** The authors declare no conflict of interest.

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
