# Peer review of "Detecting Quantum Critical Points of Correlated Systems by Quantum Convolutional Neural Network Using Data from Variational Quantum Eigensolver"

_quantumrep, doi:10.3390/quantum4040042_

Round 1
Reviewer 1 Report
Quantum machine learning is of importace. Quantum convolutional neural network (QCNN) provides a new framework with quantum circuits instead of classical neural networks as the backbone of classification methods. The authors trained the QCNN by the wavefunctions of the variational quantum eigensolver for the 1D transverse field Ising model (TFIM). The QCNN could identify wavefunctions corresponding to the paramagnetic phase and the ferromagnetic phase of the TFIM. The QCNN can be trained to predict the corresponding phase of wavefunctions around the putative quantum critical point.
It is interesting and original, I would like to suguest its acceptance after major revision.
The motivation is not clear.
The results are not explained theoretically, so More descriptions and explanations should be added to convince the audience.
The expression should be checked carefully.
Some figures are not cited correctly in the context.
The cited references do not appear correctly. The references can be updated and introduced logically if related.
Quantum convolutional neural networks. Nature Physics, 2019, 15: 1273-1278.
Quantum SUSAN edge detection based on double chains quantum genetic algorithm. Physica A, 2022: 128017.
Born machine model based on matrix product state quantum circuit. Physica A, 2022, 593: 126907.
Quantum particle swarm optimization algorithm with truncated mean stabilization strategy. Quantum Information Processing, 2022, 21(2): 42.
Author Response
We thank the referee for the generally positive assessment to our manuscript. We addressed each of his/her comments and questions in the following.
Quote: ” It is interesting and original, I would like to suggest its acceptance after major revision.”
We thank the referee for the assessment that our manuscript is interesting and original.
We have made a major revision to our manuscript to address all the comments and suggestions. The details are explained in the following.
Quote: “The motivation is not clear.”
We have added a paragraph in the introduction to highlight the motivation of our work. The major motivation is that the quantum critical point plays an important role in many strongly correlated materials, particularly the high Tc cuprates. However, the numerical simulations of such quantum critical points are often challenging and inconclusive. This motivates us to suggest using the quantum algorithm to approximate the ground state of many-body systems and then apply quantum machine learning to detect quantum phase translation.
Quote: “The results are not explained theoretically, so More descriptions and explanations should be added to convince the audience.”
We added a paragraph to describe the wavefunction being used in the VQE for the TFIM.
We added four additional paragraphs to describe the pooling units and the convolutional units of the quantum neural network.
We believe they provide more in-depth descriptions and explanations for the readers. We also provide the computer code used to generate the quantum neural network and the variational wavefunction used in this study. Interested readers can follow the details and repeat the calculations using the code provided.
Quote:”Some figures are not cited correctly in the context.”
We thank the referee for spotting this. We have fixed the order of the figures.
Quote:” The cited references do not appear correctly. The references can be updated and introduced logically if related.
Quantum convolutional neural networks. Nature Physics, 2019, 15: 1273-1278.
Quantum SUSAN edge detection based on double chains quantum genetic algorithm. Physica A, 2022: 128017.
Born machine model based on matrix product state quantum circuit. Physica A, 2022, 593: 126907.
Quantum particle swarm optimization algorithm with truncated mean stabilization strategy. Quantum Information Processing, 2022, 21(2): 42.”
We thank the referee for the comment. We checked the references and fixed any possible typos or errors. We thank the referee for pointing out these references. We have included them in the references list of our revised manuscript.
Reviewer 2 Report
The authors applied a Quantum Convolutional Neural Network as a Classifier for Many-Body Wavefunctions from the Quantum Variational Eigensolver. The state of art , the method and the argument of the results are well described, I suggest the publication of this work strongly, just please give an instruction of the N-ZZ Ising coupling gates, because this type of gate is not so popular. And please add the Cirq code.Author Response
We thank the referee for the positive comments on our manuscript. We added an additional description of the N-ZZ Ising coupling gates in the caption of the figure 1. We also include the Cirq code in the supplementary materials. Interested readers can use the code to reproduce results and modify the code to generate new results.
Reviewer 3 Report
The development of efficient methods of quantum computations is very important at the present time because of a large variety of problems requiring such an approach. On the other hand, many platforms have become recently available for implementation. In this sense, the proposed application of a quantum convolutional neural network for the determination of critical points and subsequent studies of their vicinities is timely and significant. The authors made an important first step in this direction, and I believe that the present manuscript merits publication in Quantum Reports.
I have the following question and suggestions to further improve the manuscript:
- For actual calculations, the authors used 81 points available on Tensorflow Quantum with the linear interpolation between the points. Would be the results more reliable if a larger quantum computer like the D-Wave machine is used?
- Naively, I expect an improvement in performance with increasing the number of layers. However, the most accurate prediction is for N=8, not N=12. Can authors comment on that?
- I would suggest additional careful reading of the manuscript to eliminate stylistic and grammatical errors. In particular, J=1 as the energy scale is introduced twice in lines 168 and 174, and the grammar in lines 346-348 is flawed.
Author Response
We thank the referee for the positive assessment of our manuscript. We address the specific comments and questions in the following.
Quote:” For actual calculations, the authors used 81 points available on Tensorflow Quantum with the linear interpolation between the points. Would be the results more reliable if a larger quantum computer like the D-Wave machine is used?”
Since it is a machine-learning based approach. The larger the available database, the higher chance of training the quantum neural network with better accuracy. We expect a larger number of data points would improve the estimation in the present study too.
Quote:” Naively, I expect an improvement in performance with increasing the number of layers. However, the most accurate prediction is for N=8, not N=12. Can authors comment on that?”
The expectation is correct that a larger system size should represent a better estimation of the quantum critical point. The fact that the result for N=8 is closer to that of the N=12 may be due to the accuracy of the training being affected by the increase in the system size. The larger quantum neural network being used for N=12 may not be optimized as well as that for N=8.
Quote:” I would suggest additional careful reading of the manuscript to eliminate stylistic and grammatical errors. In particular, J=1 as the energy scale is introduced twice in lines 168 and 174, and the grammar in lines 346-348 is flawed.”
We have proofread the manuscript again and fixed stylistic and grammatical errors. We thank the referee for pointing these out. In particular, lines 168-174 and 346-348 are corrected.
Round 2
Reviewer 1 Report
The title is too long and it should be shortened.
The new progresscan be covered.
Signal Processing: Image Communication, 2022, 116891. https://doi.org/10.1016/j.image.2022.116891.
The English can be improved.
Author Response
We thank the referee for the second report.
We addressed the comments as follows.
1. We changed the title to "Detecting Quantum Critical Points of Correlated Systems by Quantum Convolutional Neural Network using data from
Variational Quantum Eigensolver"
2. We added the reference, Signal Processing: Image Communication, 2022, 116891. https://doi.org/10.1016/j.image.2022.116891.
3. We proofread the manuscript and made some minor edits in the text.
We believe the revised manuscript is fit to publish in the Quantum Reports.